# Grain engineering for efficient near-infrared perovskite light-emitting diodes

Sung-Doo Baek [1,16], Wenhao Shao [1,16], Weijie Feng[2], Yuanhao Tang[1], Yoon Ho Lee[1], James Loy[3], William B. Gunnarsson[4], Hanjun Yang[1,5], Yuchen Zhang[6], M. Bilal Faheem[6], Poojan Indrajeet Kaswekar[6], Harindi R. Atapattu[7], Jiajun Qin[8], Aidan H. Coffey[9], Jee Yung Park[1,10,11], Seok Joo Yang[1], Yu-Ting Yang[1], Chenhui Zhu[9], Kang Wang[1,12], Kenneth R. Graham[7], Feng Gao[8], Quinn Qiao[6], L. Jay Guo[2,13], Barry P. Rand[4,14] & Letian Dou[1,5,15]✉

Metal halide perovskites show promise for next-generation light-emitting diodes, particularly in the near-infrared range, where they outperform organic and quantum-dot counterparts. However, they still fall short of costly III-V semiconductor devices, which achieve external quantum efficiencies above 30% with high brightness. Among several factors, controlling grain growth and nanoscale morphology is crucial for further enhancing device performance. This study presents a grain engineering methodology that combines solvent engineering and heterostructure construction to improve light outcoupling efficiency and defect passivation. Solvent engineering enables precise control over grain size and distribution, increasing light outcoupling to ~40%. Constructing 2D/3D heterostructures with a conjugated cation reduces defect densities and accelerates radiative recombination. The resulting near-infrared perovskite light-emitting diodes achieve a peak external quantum efficiency of 31.4% and demonstrate a maximum brightness of 929 W sr$^{-1}$ m$^{-2}$. These findings indicate that perovskite light-emitting diodes have potential as cost-effective, high-performance near-infrared light sources for practical applications.

Efficient perovskite light-emitting diodes (PeLEDs) necessitate delicate grain engineering. Such crystallization control over polycrystalline perovskite films not only implies effective defect passivation to enable efficient charge recombination but extends externally to the film morphology and grain distribution, which significantly impact light extraction[1]. A study in 2018 reported the spontaneous formation of discrete perovskite islands on a ZnO transport layer, which gave rise to an exceptional light outcoupling efficiency (OCE) of ~30%. Together

[1]Davidson School of Chemical Engineering, Purdue University, West Lafayette, IN, USA. [2]Macromolecular Science and Engineering, University of Michigan, Ann Arbor, MI, USA. [3]Department of Physics, Princeton University, Princeton, NJ, USA. [4]Department of Electrical and Computer Engineering, Princeton University, Princeton, NJ, USA. [5]Department of Chemistry, Purdue University, West Lafayette, IN, USA. [6]Department of Mechanical and Aerospace Engineering, Syracuse University, Syracuse, NY, USA. [7]Department of Chemistry, University of Kentucky, Lexington, KY, USA. [8]Department of Physics, Chemistry and Biology (IFM), Linköping University, Linköping, Sweden. [9]Advanced Light Source, Lawrence Berkeley National Laboratory, Berkeley, CA, USA. [10]Department of Chemical and Environmental Engineering, Yale University, New Haven, CT, USA. [11]Energy Sciences Institute, Yale University, West Haven, CT, USA. [12]Key Laboratory of Photochemistry, Institute of Chemistry, Chinese Academy of Sciences, Beijing, China. [13]Department of Electrical Engineering and Computer Science, University of Michigan, Ann Arbor, MI, USA. [14]Andlinger Center for Energy and the Environment, Princeton University, Princeton, NJ, USA. [15]Birck Nanotechnology Center, Purdue University, West Lafayette, IN, USA. [16]These authors contributed equally: Sung-Doo Baek, Wenhao Shao. ✉e-mail: dou10@purdue.edu

with the introduction of amino acid additives for passivating defects, near-infrared (NIR) PeLEDs with 20% external quantum efficiency (EQE) were demonstrated[2]. Numerous attempts building on this approach have been made since to elevate device efficiencies[3–7]. However, these efforts often rely mainly on additive engineering and lack a thorough, innovative focus on grain engineering and morphological control that balances external and internal factors.

In this work, we demonstrate a holistic grain engineering approach to modulate the extrinsic light extraction and intrinsic defect passivation for highly efficient NIR PeLEDs based on cubic-phase formamidinium lead triiodide (α-FAPbI₃). These advancements culminate in significantly enhanced device performance, as evidenced by a peak EQE of 31.4% (average 26.9%) and a maximum radiance of 929 W sr⁻¹m⁻² (observed across different devices). Via solvent engineering, we achieve an optimized discrete island−convex dome morphology that promoted OCE to approximately 40%. Subsequently, band-alignment-tailored 2D/3D heterostructures are constructed on the as-grown perovskite grains, preserving the optimized film morphology, using a narrow-bandgap organic spacer named TeFBTT, which stands for 2-(5-(7-(3-ethylthiophen-2-yl)-5,6-difluorobenzo[c][1,2,5]thiadiazol-4-yl)thiophen-2-yl)ethan-1-aminium iodide. The narrow-gap TeFBTT features close Type-I band alignment in layered perovskite phases, ensuring efficient energy transfer. The construction of 2D/3D heterostructure by the post-treatment of TeFBTT significantly enhances defect passivation, resulting in a greatly improved photoluminescence quantum yield (PLQY).

## Results and discussion

### Grain size and distribution control via solvent engineering

In our solvent engineering approach, a mixture of *N,N*-dimethylformamide (DMF) and *N*-methylpyrrolidone (NMP) was used to dissolve the perovskite precursors. Prior studies in the field of perovskite solar cell research have demonstrated that the addition of NMP in DMF can yield significantly more stable, thick, and continuous FAPbI₃ films with fewer defects, achieved through the formation of a stable intermediate phase that facilitates the control of nucleation and crystal growth[8,9]. However, it is noteworthy that this approach is rarely explored in the context of PeLEDs and the effects of co-solvent on the grain growth of a thin light-emitting film remain unknown. Figure 1a displays the morphology of polycrystalline FAPbI₃ films obtained with gradually increased NMP content in the precursor solution, examined by scanning electron microscopy (SEM) and atomic force microscopy (AFM) (Supplementary Fig. 1). It is generally observed that amino acid additives−5-aminovaleric acid (5AVA) in this case−can easily lead to the formation of discontinuous FAPbI₃ islands on a zinc oxide (ZnO)/polyethylenimine ethoxylated (PEIE) layer, primarily due to a dehydration reaction of 5AVA with ZnO-PEIE surface during annealing[2]. This reaction facilitates the formation of a FAPbI₃ island structure while concurrently generating thin organic layers that bridge voids between the discontinuous grains, thereby mitigating shunt current pathways. This phenomenon becomes more pronounced with the addition of NMP, primarily attributed to its discernibly higher boiling point compared with DMF, resulting in larger crystal grains. Therefore, compared with the control sample fabricated with a pure DMF solution, increasing the NMP ratio in DMF through 14:1 to 10:1 led to gradually increased average grain size along with decreased packing density. However, crystals became more aggregated when an excess amount of NMP was introduced, reaching a ratio of 2:1 (DMF:NMP), as shown in Fig. 1a. To investigate the influence of NMP on crystal phase and quality, X-ray diffraction (XRD) and photoluminescence (PL) analyses were performed (Supplementary Figs. 2a, b). All films exhibited strong diffraction peaks from α-FAPbI₃ without impurity phases except for the condition of DMF:NMP = 2:1, where δ-FAPbI₃ and PbI₂ are evident. This implies that an excessive amount of NMP, possessing a high boiling point, hinders the crystallization toward the desired α-FAPbI₃, which in turn leads to weak desired NIR emission and multiple impurity

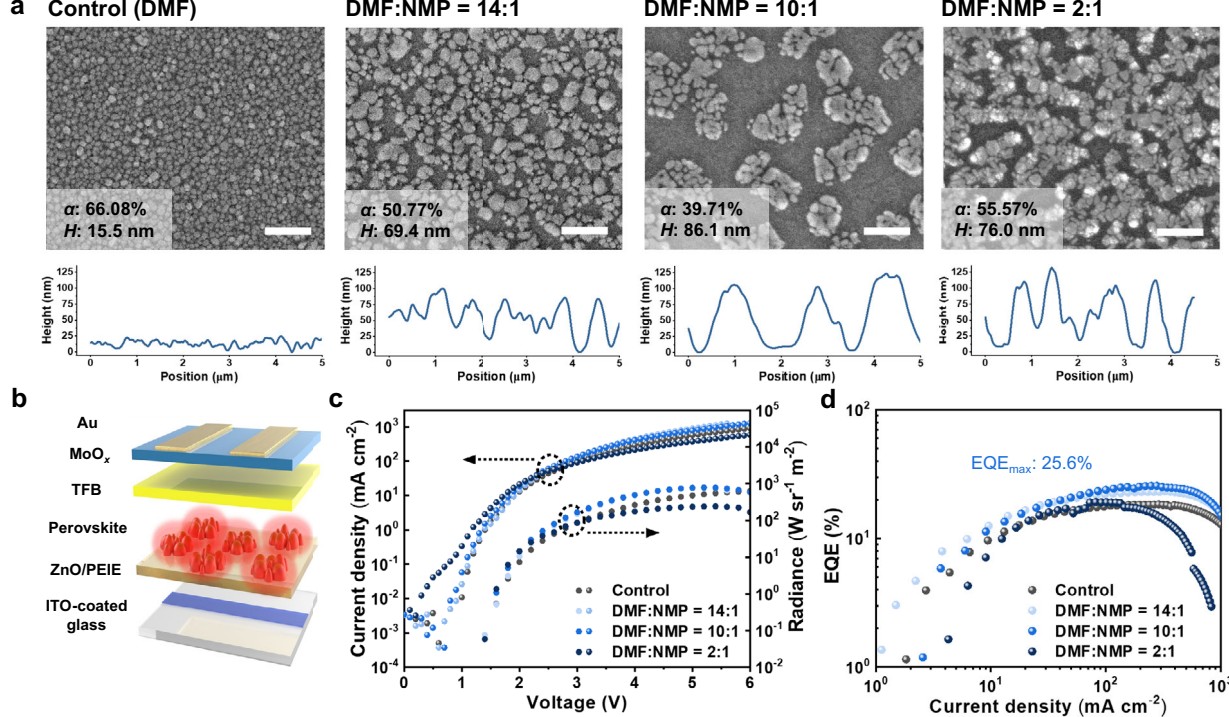

**Fig. 1 | Solvent-Engineering Approach to PeLEDs. a** SEM images and AFM height profiles of perovskite films with varying solvent mixing ratios (scale bar: 1 μm, *α*: packing density, *H*: average height). **b** Device structure, **c** current density−voltage−radiance (*J*−*V*−*R*) curves, and **d** EQE characteristics of PeLEDs based on perovskite films with varying solvent mixing ratios.

PL peaks originating from defect states in δ-FAPbI$_3$ and/or PbI$_2$ phases[10,11] (Supplementary Fig. 2a). The appropriate amount of NMP, on the other hand, can lead to enhanced PL and a narrower full width at half maximum (FWHM) compared to the use of pure DMF (FWHM ca. 45 nm). Specifically, the DMF:NMP = 10:1 solution produced perovskites that exhibited the highest PL intensity at 800 nm with a narrower FWHM (~41 nm).

We fabricated LEDs based on these perovskite films with the device structure of indium tin oxide (ITO) (120 nm)/ZnO-PEIE (25 nm)/perovskite/TFB (30–50 nm)/MoO$_x$ (10 nm)/Au (60 nm) (Fig. 1b, Supplementary Fig. 3). The performance of PeLEDs with varying solvent mixing ratio was compared in Fig. 1c,d. Remarkably, devices with the optimum solvent mixing ratio of DMF:NMP = 10:1 produced a maximum radiance of 787 W sr$^{-1}$ m$^{-2}$ and a maximum EQE of 25.6% at 798 nm (Supplementary Fig. 2c), which are significantly higher than those of the control device (radiance$_{max}$: 630 W sr$^{-1}$ m$^{-2}$, EQE$_{max}$: 18.4%, peak electroluminescence, EL: 795 nm). It is particularly noteworthy that EQE reached the peak value of 25.6% at a high current density of 278 mA cm$^{-2}$ and remained over 15% even at 1000 mA cm$^{-2}$. For other devices with mixed solvent ratios of 14:1 and 2:1, maximum EQEs were measured to be 23.3% and 19.1%, respectively.

## Optimized discrete island–convex dome structure

One standout advantage of discontinuous films is their exceptional OCE[1,2]. Randomly distributed perovskite crystals act as discrete emitters and scatterers, which when combined with sequentially deposited layers, form convex structures. Due to the high reflectivity of gold, these convex domes effectively function as microlenses to facilitate the extraction of light that would otherwise be confined in waveguide modes[2]. This discrete island–convex dome morphology was optimized with the addition of NMP, resulting in dramatically increased average grain size and notably reduced packing density (Fig. 1a, Supplementary Fig. 1, and Supplementary Fig. 4), as validated with cross-sectional transmission electron microscopy (TEM) (Supplementary Fig. 3).

The impact of morphology on the OCE of our devices was modeled progressively with 3D finite-difference-time-domain (3D-FDTD) simulations. First, the effect of convex domes was isolated with a single-cell model to eliminate the effects from nearby grains. A sudden OCE jump from 5% to 25% was observed from continuous films to discrete islands with no convex domes (i.e., convex height, $h_s = 0$). Subsequently, the OCE gradually increased with $h_s$ (Fig. 2a, b). Building on this initial observation, we combined convex domes with discrete island morphology and further calculated the OCE based on specific perovskite film morphologies using a simplified periodic grain distribution (a method adopted from the literature[2]; refer to Methods for more details). From the control condition through a DMF:NMP mixing ratio of 14:1 to 10:1, the simulated OCEs were significantly boosted from 26.1%, 32.8%, to 40.4%, respectively (Fig. 2c). Notably, the high OCE of 40.4% surpasses the previously reported discrete-island FAPbI$_3$ PeLED (~30%)[2]. This remarkable enhancement arises from a combination of convex domes, increased grain size (represented by its height, $H$), and decreased packing density ($\alpha$), all driven by the addition of NMP. The effects of the latter two were also studied under the periodic-distribution model (Fig. 2c). Results pointed to an optimum $\alpha$ and $H$ near 40% and 86 nm, respectively, which serendipitously aligned with the characteristics extracted from the DMF:NMP = 10:1 film. In other words, through deliberate solvent engineering with NMP addition, the grain size and packing density of a polycrystalline perovskite film can be tailored to have the optimum light scattering behavior, facilitating efficient light outcoupling.

Due to the modulated discrete island – convex dome structure, light extraction from each grain became relatively localized, facilitated mostly by their own microlens and no longer dependent on nearby structures over a long distance. This characteristic stands in contrast to a continuous film model where emission can be efficiently waveguided. Therefore, concerning light extraction, the simplified model with periodically distributed grains may not differ significantly from the actual film morphology where grains are randomly distributed and clustered. To investigate this further, we modeled the actual grain distribution and even considered size variation (Fig. 2d, e). As anticipated, the OCE remained stable near 40%.

## Tailoring 2D/3D heterostructures

Although the high OCE is critical for LED performance, it is equally essential to enhance radiative carrier recombination in perovskite films. To further boost device performance while preserving the optimized crystal morphology, we designed a narrow-bandgap organic cation, TeFBTT, for post-treatment. This organic spacer is capable of forming layered 2D perovskites with a properly tailored Type-I band alignment that ensures efficient energy transfer to inorganic motifs, which is critical in PeLEDs[12]. The molecular structure of TeFBTT and the crystal structure of the resultant 2D perovskite are illustrated in Fig. 3a. In detail, the fluorinated benzothiadiazole (FBT) core endows TeFBTT with deep highest occupied and lowest unoccupied molecular orbitals (HOMO and LUMO) to ensure close alignment with the valence and conduction band of [PbI$_4$]$^{2-}$ layer, respectively, as predicted from density functional theory (DFT) and further supported by experimental energy level determination (Supplementary Figs. 5a–d). To prevent self-aggregation of organic spacers and facilitate layered perovskites formation, a flexible ethyl side chain was attached to the terminal thiophene to increase solubility and simultaneously twist the backbone from a completely planar structure. The crystal structure of (TeFBTT)$_2$PbI$_4$ revealed well-aligned TeFBTT spacers with a pronounced torsion angle of 52° between FBT and the terminal thiophene (Fig. 3a). These characteristics culminate in the facile fabrication of (TeFBTT)$_2$PbI$_4$ thin films with a well-defined XRD pattern resembling layered perovskites and strong PL emission with a single sharp excitonic peak at 532 nm, confirming the Type-I alignment (Supplementary Figs. 5e, f).

TeFBTT was used to post-treat the solvent-engineered perovskite films (DMF:NMP = 10:1). The spontaneous formation of 2D/3D heterostructures (Fig. 3b) was verified with grazing incidence wide-angle X-ray scattering (GIWAXS) patterns of the post-treated FAPbI$_3$ film (Fig. 3c, Supplementary Fig. 6), which revealed the presence of layered perovskite phases. The impact of the 2D/3D heterostructure on PeLEDs was subsequently investigated with the aforementioned device structure (Fig. 1b, Supplementary Fig. 7). Given that the quantity of 2D phase affects the resulting device performance, we first optimized the concentration of TeFBTT in an isopropanol (IPA)/chlorobenzene solution used for post-treatment (Fig. 3d, Supplementary Fig. 8). The optimal device performance was achieved with a concentration of 0.02% w/v, where the discontinuous polycrystalline film retained its pre-treatment morphology (Supplementary Fig. 9, Fig. 1a). The approximate 2D/3D phase ratio of the optimal device was estimated to be 0.5% by comparing 1D GIWAXS pattern (Supplementary Fig. 10). Additionally, both the PL (Supplementary Fig. 9b) and EL (Fig. 3e) spectra exhibited minimal shifts with and without TeFBTT treatment at the optimized concentration.

The peak EQE was enhanced up to 31.4% at a current density of 62.7 mA cm$^{-2}$ following the TeFBTT treatment with a maximum radiance of 784 W sr$^{-1}$ m$^{-2}$ (Fig. 3f, g). This is in direct contrast to the poor device performances when wide bandgap organic cations were used without band alignment consideration (Supplementary Fig. 11). Additionally, our PeLEDs exhibited high reproducibility as shown in Fig. 3h, where the peak EQE histograms displayed mean value of 16.5%, 23.7%, and 26.9%, with standard deviations of 1.1%, 1.0%, and 1.6% for control, solvent-engineered, and TeFBTT-treated devices, respectively. Apart from the champion device achieving a peak EQE of 31.4% and a maximum radiance of 784 W sr$^{-1}$ m$^{-2}$, other representative devices exhibited peak EQEs of 29.6% and 26.1%, along with impressive maximum

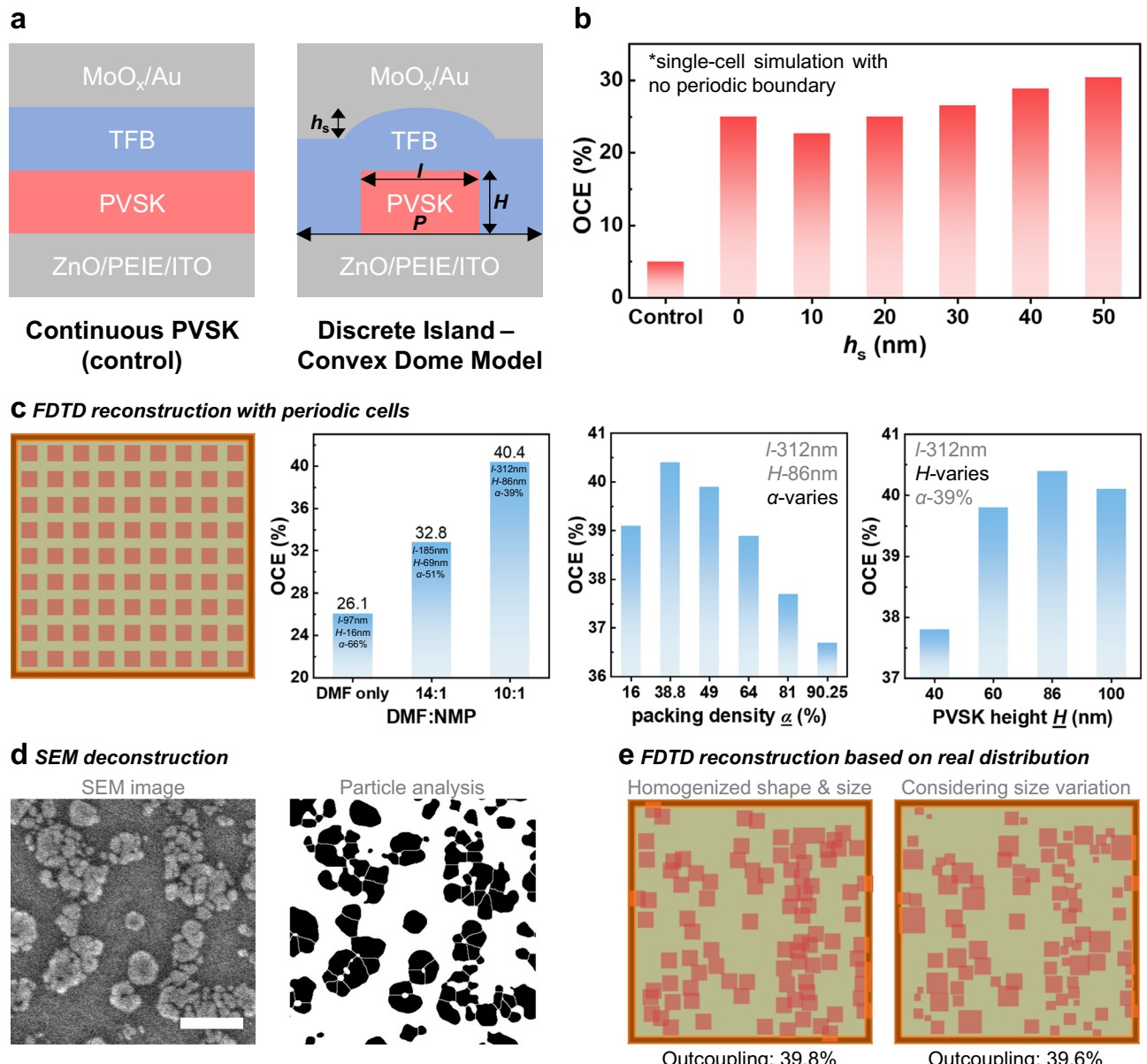

**Fig. 2 | Outcoupling efficiency enhancement by the solvent-engineering approach. a** Device and perovskite (PVSK) film morphology were modeled in 3D-FDTD simulation. A single cell (size: *P*) contains one perovskite grain simplified as a tetragonal block embedded in TFB with defined length (*l*) and height (*H*). The curvature atop TFB was simplified as a convex dome defined by the convex height ($h_s$). Due to the high reflectivity of MoO$_x$/Au, each convex dome serves as an intrinsic lens to focus the perovskite emission and prevent unwanted scattering, thus enhancing the total OCE. **b** To model this effect, a single-cell simulation was

carried out while varying the convex geometry $h_s$. The alternative model based on conventional continuous perovskite layer was also studied as the control. An initial quantum OCE jump was observed from continuous film to discrete islands. **c** The single-cell was expanded to a 9 × 9 periodic grain distribution to estimate the real-scale OCE while modeling the effects from solvent mixture, packing density (*α*), and *H*. **d** Demonstration of the SEM deconstruction process to obtain grain distribution parameters such as *P* and *α* as well as to extract real grain distribution, which was then modeled in **e** to estimate the real device OCEs (scale bar: 1 μm).

radiances of 825 and 929 W sr$^{-1}$ m$^{-2}$, respectively (Supplementary Figs. 12, 13, and Supplementary Table 1). The device stability was assessed at different current densities (Fig. 3i), revealing $T_{50}$ values of 73, 21, and 9.3 h at 20, 50, and 100 mA cm$^{-2}$, respectively.

### Effective defect passivation
Besides morphological factors, the mixed solvent of DMF and NMP offers an advantage in suppressing defect formation through intermediate phase-assisted crystallization[8,9] (Fig. 4a). The optimal solvent-engineered film consistently exhibited higher PLQY irrespective of laser power compared to the control film, implying the presence of fewer defects. Notably, a significant PLQY increase was observed after

TeFBTT treatment, with the highest observed value reaching 82.4%, indicating effective defect passivation with the construction of the 2D/3D heterostructure. Given that the maximum OCE of our device was calculated to be 40.4%, the theoretical EQE could reach approximately 33.3%. This suggests that our measured EQE of 31.4% falls within a reasonable range.

In addition, nanoscale charge-carrier recombination lifetime mapping on each grain provided another perspective to visualize the carrier dynamics as the electrical injection is considered during device operation[13] (Fig. 4b, Supplementary Fig. 14). The average charge-carrier recombination lifetimes for control, solvent-engineered, and TeFBTT-treated films were measured to be 12.0, 9.2, and 8.8 μs,

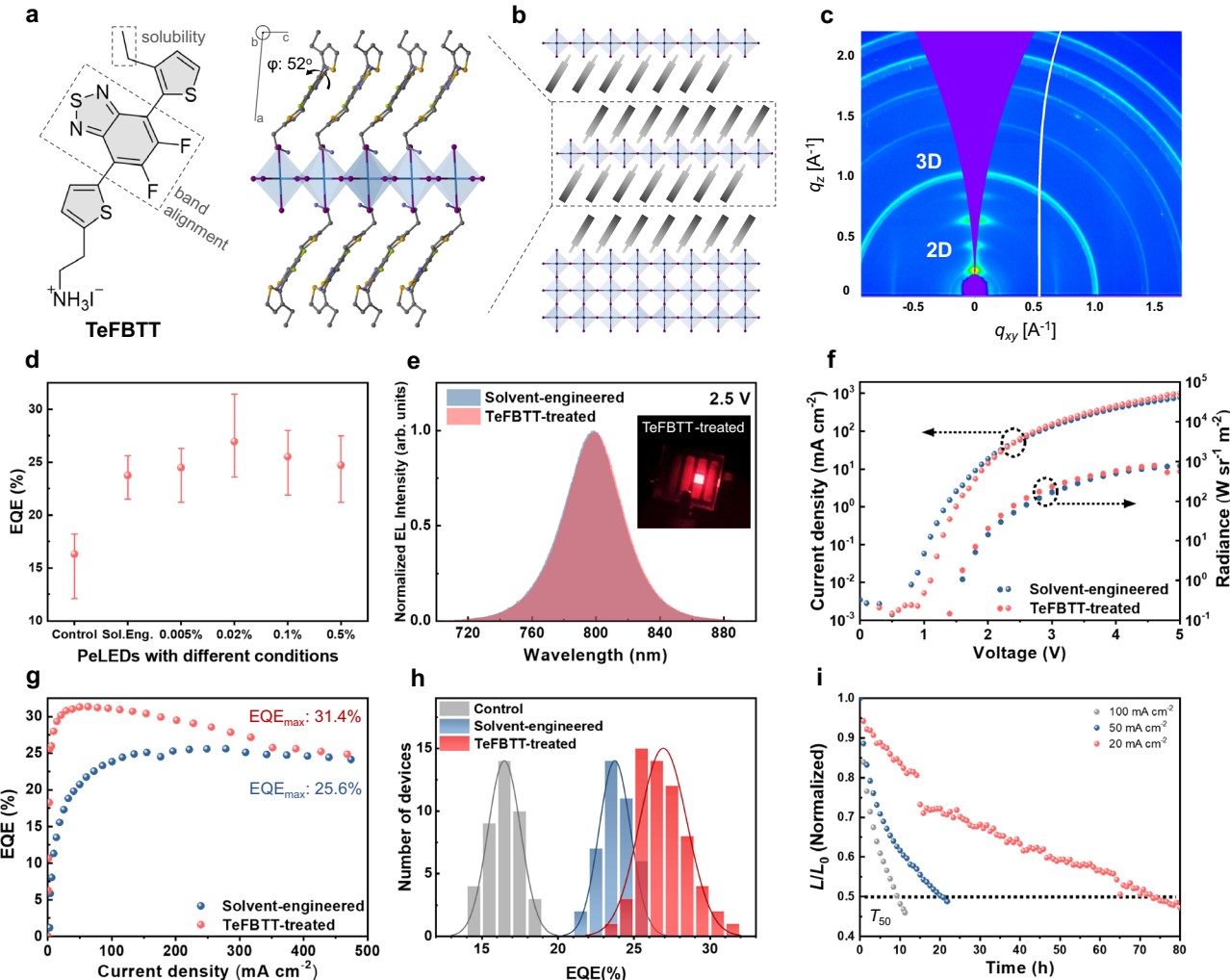

**Fig. 3 | The 2D/3D Perovskite Heterostructure Approach to PeLEDs. a** Chemical structure of TeFBTT and single crystal structure of (TeFBTT)$_2$PbI$_4$. **b** Schematic representation of the TeFBTT-incorporated 2D/3D perovskite heterostructure. **c** GIWAXS pattern of a perovskite film with TeFBTT post-treatment. **d** EQEs of PeLEDs under different conditions (error bars indicate minimum and maximum

EQEs). **e** EL spectra, light emission camera image (inset), **f** J–V–R curves, and **g** EQE characteristics of PeLEDs with solvent-engineered and TeFBTT-treated perovskite films. **h** Histograms depicting EQE$_{max}$ of PeLEDs. **i** The LED stability test for the optimized TeFBTT-treated device under different current densities.

respectively. This shows a gradually accelerated radiative recombination after solvent engineering and TeFBTT treatment. We attribute this trend to the passivation of shallow defects (Supplementary Fig. 15), which is consistent with the observed enhancement in device performance, as faster radiative recombination is generally preferred in LED operation[7]. The mapping images reveal that the solvent-engineered film exhibits a marginally reduced recombination rate at its periphery. Conversely, in the case of TeFBTT-treated film, the recombination rate displays a more uniform distribution across the entire region, indicating a uniform organic cation coating on the surface of perovskites, yielding a higher quality film.

## Device cross-validation

While we were able to successfully fabricate and test highly performing PeLEDs, cross-validating the device performances with other facilities establishes their reproducibility and reliability. Furthermore, shipping samples between labs supports the notion of a device being reliable given the extended time between fabrication and characterization as well as the environmental stresses of packaging and transport. Since a widely accepted certification agency has yet to be established, unlike in the solar cell field, our devices were systematically cross-validated with another group at Princeton University. Supplementary Fig. 16a

illustrates a scheme for angle-integrated EQE measurement based on a gonio-photodetector, used at Princeton University, which enables the collection of angular distribution data for emitted light from the devices (Supplementary Fig. 17). Notably, larger substrates (3 × 3 cm$^2$) were adopted with an increased active area of 0.1 cm$^2$. The device fabrication process was carried out at Purdue University under identical optimized conditions, and then devices were carefully sealed and shipped to Princeton. Supplementary Figs. 16b, c present the characteristics of the fabricated LEDs as measured at Princeton. The J–V curve exhibited similarity (Supplementary Fig. 16e), and the radiance was measured to be 592 W sr$^{-1}$ m$^{-2}$, falling within a reasonable radiance range of in-house performance. The peak EQE was recorded as 26.5%, which is close to the average EQE from our original device distribution, supporting the accuracy of our measurements.

It is worth noting that device performance may exhibit slight variations from batch to batch, despite stringent fabrication conditions being maintained each time. Consequently, while the batch of devices were prepared for cross-validation at Princeton, we also concurrently fabricated a device under precisely the same conditions to conduct a second validation at Purdue University. The device (1.4 cm × 1.6 cm substrate, active area: 0.04 cm$^2$) was assessed using an integrating sphere (Supplementary Fig. 16d) and its

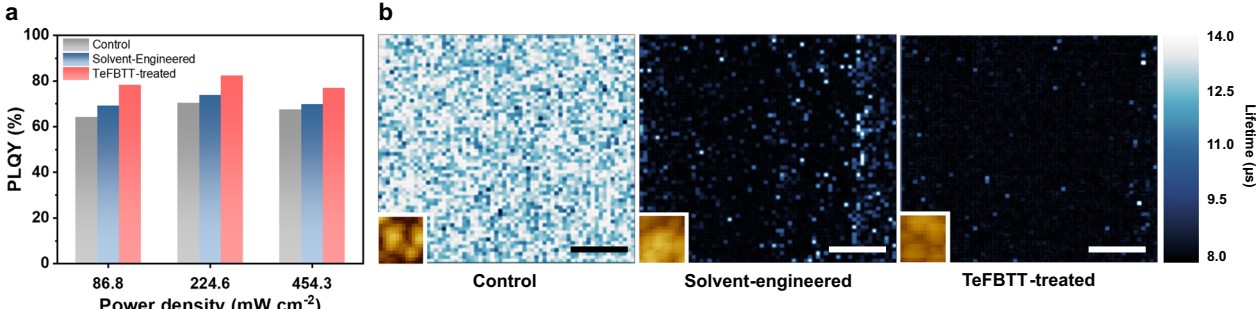

**Fig. 4 | Effective Defect Passivation by the 2D/3D Perovskite Heterostructure Approach.** (**a**) Maximum PLQY and (**b**) nanoscale charge-carrier recombination lifetime mapping and corresponding AFM images (inset) of perovskite films with control, solvent-engineered, and TeFBTT-treated condition (scale bar: 20 nm).

characteristics are outlined in Supplementary Fig. 16e, f. The $J$–$V$–$R$ trends were nearly identical in operational range, turn-on voltage, and curve shape between the two cross-validation methods. The peak radiance and peak EQE were measured at 500 W sr$^{-1}$ cm$^{-2}$ and 26.8%, respectively, closely mirroring those obtained from the cross-validation institution. These results conclusively and unambiguously demonstrate that our device performance has been accurately assessed and that the device is not only reproducible (even on larger substrates) but also transferable without noticeable degradation for further use at other institutions.

In this study, we perform comprehensive extrinsic/intrinsic grain engineering by combining two simple and synergistic but previously overlooked strategies—solvent engineering and 2D/3D heterostructures. The performance of NIR PeLEDs are significantly enhanced, achieving a peak EQE of 31.4% through a morphologically modulated discrete island—convex dome structure via solvent mixtures, and through band-alignment-tailored perovskite 2D/3D heterostructures based on semiconducting TeFBTT spacers. Systematic optical modeling and photophysical studies reveal the underlying factors as a combined effect of outcoupling modulation and defect passivation. This holistic strategy proves highly reproducible not only internally from batch to batch but also through external cross-validation across institutions. This work not only establishes a robust methodology for optimizing NIR PeLEDs but also provides valuable insights into the fundamental factors influencing their performance. The superior characteristics demonstrated by our devices, coupled with their high reproducibility, position them as promising contenders for various applications in optoelectronics.

## Methods
### Materials
The perovskite precursor solutions were prepared by following previously reported literature with slight modifications[2]. FAI (99.99%, Greatcell Solar Materials), PbI$_2$ (99.99%, TCI America), and 5AVA (97%, Sigma Aldrich) with a molar ratio of 2.5:1:0.7 were dissolved in mixed solvents of DMF and NMP (anhydrous, Sigma Aldrich) with different ratios at 60 °C for 30 min while stirring (The solution chemistry between 5AVA and FAI is discussed in Supplementary Note 1, along with Supplementary Figs. 18–21). For device stability tests, the 5AVA ratio was increased to 1.0. Subsequently, the mixture was stirred overnight without additional heating. The total precursor concentrations of the resulting solutions were maintained at 10 wt%. PEIE (80% ethoxylated solution), TFB, and MoO$_3$ (99.97%) were purchased from Sigma Aldrich. Regarding perovskite single crystal growth, PbI$_2$ (99.999% trace metals basis, perovskite grade), hydroiodic acid (HI, 57 wt.% in H$_2$O), and hypophosphorous acid (H$_3$PO$_2$, 50 wt.% in H$_2$O) were purchased from Sigma Aldrich. Regarding organic synthesis, deuterated dimethyl sulfoxide (DMSO-d$_6$) and deuterated chloroform (CDCl$_3$) were purchased from Sigma Aldrich. Reagents and precursors used in organic synthesis were purchased from Sigma Aldrich or Fisher Scientific.

### Synthesis of ZnO nanocrystals
ZnO nanocrystals (NCs) were prepared following a previously reported method[14]. A 0.1 M solution of zinc acetate dihydrate was prepared by dissolving the compound in 50 mL of DMSO. After complete dissolution, 10 mL of a 0.5 M tetramethylammonium hydroxide solution in ethanol was slowly added over 1 min. The mixture was stirred and reacted at 30 °C for 1 h. The ZnO NCs were then precipitated by adding an excess of acetone, followed by centrifugation at 3000 rpm for 15 min. Finally, the ZnO NCs were resuspended in ethanol at a concentration of 14 mg mL$^{-1}$ for LED device fabrication.

### Synthesis of TeFBTT
The complete synthetic scheme is included in Supplementary Fig. 22.

*tert*-butyl (2-(5-(tributylstannyl)thiophen-2-yl)ethyl)carbamate was synthesized as previously reported[15]. *tert*-butyl (2-(thiophen-2-yl)ethyl) carbamate (1 equiv.) was dissolved in anhydrous THF (0.2 M) and a 1.6 M solution of n-BuLi in hexane (2.5 equiv.) was added dropwise at 0 °C. The mixture was stirred for 3 h at 0 °C and then trimethyltin chloride (2.5 equiv.) was added. The temperature was raised to room temperature and the solution was stirred for 2 h. Afterwards the solution mixture was poured into water and extracted with DCM. The organic phase was washed with water and brine and dried over Mg$_2$SO$_4$. The solvent was removed under reduced pressure to produce yellowish oil and directly used for the next step without further purification.

*2-bromo-3-ethylthiophene*: A flame-dried round-bottom flask was equipped with a magnetic stir bar and a dropping funnel. A solution of 3-ethylthiophene (1.1219 g, 10 mmol, 1 equiv.) in anhydrous acetonitrile (10 mL, 1 M) was added to the flask. A solution of N-bromo-succinimide (1.958 g, 11 mmol, 1.1 eq) in anhydrous acetonitrile (22 mL, 0.5 M) was added to the dropping funnel. The entire system was sealed and protected with argon, after which the solution containing 3-ethylthiophene was heated to 40 °C. While vigorously stirring, the NBS solution was added dropwise to the flask with a dropping speed of 1 drop min$^{-1}$ (syringe pump is recommended for future study). The procedure was designed to maintain a low NBS concentration at any given time and thus, prevent the di-substitution of 3-ethylthiophene with bromine. After the addition of NBS, stirring was continued for another hour before the solvent was evaporated with a rotary evaporator. The residue was dissolved in EA, washed twice with 2 M aq. NaOH followed by brine, dried over MgSO$_4$, filtered, and concentrated. The product (58.4% yield) was used without further purification. Less than 5 mol% of the starting material, 3-ethylthiophene was detected from the product through nuclear magnetic resonance (NMR), while no di-substituted product was detected. $^1$H NMR (400 MHz, CDCl$_3$) δ 7.19 (d, J = 5.6 Hz, 1H), 6.82 (d, J = 5.6 Hz, 1H), 2.59 (q, J = 7.6 Hz, 2H), 1.19 (t, J = 7.6 Hz, 3H).

*tributyl(3-ethylthiophen-2-yl)stannane*: To a solution of 2-bromo-3-ethylthiophene (1.1161 g, 5.8407 mmol, 1 equiv.) in ether (11.68 mL, 2 mL mmol$^{-1}$) at −78 °C was added n-BuLi (2.5 M in hexane, 2.57 mL,

6.4248 mmol, 1.1 equiv.) slowly dropwise. The solution gradually turned yellow, and the reaction was kept stirring at −78 °C for 80 min. Then, tributylstannyl chloride (1.74 mL, 6.4248 mmol, 1.1 equiv.) was added at the same temperature. The mixture immediately turned colorless and was allowed to warm to room temperature naturally and kept stirring for 12 h. After washing with water, the organic phase was collected, and the aqueous phase was washed with ether. The combined organic phases were washed with brine, dried, and the solvent was removed through rotary evaporation to provide the desired product (2.9663 g) as a clear pale-yellow oil which was used without further purification. The product is assumed to be 79% pure by assuming a 100% yield.

*BrFBTT-Boc, tert-butyl (2-(5-(7-bromo-5,6-difluorobenzo[c][1,2,5]thiadiazol-4-yl)thiophen-2-yl)ethyl)carbamate*: *tert*-butyl (2-(5-(trimethylstannyl)thiophen-2-yl)ethyl)carbamate (1.951 g, 5 mmol), tris(dibenzylideneacetone)dipalladium(0) (37 mg, 2 mol%), tri(o-tolyl) phosphine (49 mg, 8 mol%), and 4,7-dibromo-5,6-difluorobenzo[c] [1,2,5]thiadiazole (3.299 g, 10 mmol) were mixed in a 100 mL flame-dried round bottom flask. After argon protection, anhydrous toluene (50 mL, 0.1 M) was added via syringe. The mixture was refluxed for 12 h at 105 °C. After cooling to room temperature, water was added, and the mixture was extracted with dichloromethane (DCM). The organic layers were combined, washed with brine and dried over sodium sulfate. The solids were removed by filtration, solvents were removed under vacuum and the residue was purified through silica column chromatography (hexane:ethylacetate). Desired mono-substituted product was isolated with 28.8% yield. $^1$H NMR (400 MHz, CDCl$_3$) δ 8.11 (d, J = 3.8 Hz, 1H), 7.00 (d, J = 3.9 Hz, 1H), 3.48 (d, J = 4.4 Hz, 2H), 3.12 (t, J = 6.7 Hz, 2H), 1.45 (s, 9H).

*TeFBTT-Boc, tert-butyl (2-(5-(7-(3-ethylthiophen-2-yl)-5,6-difluorobenzo[c][1,2,5]thiadiazol-4-yl)thiophen-2-yl)ethyl)carbamate*: tributyl(3-ethylthiophen-2-yl)stannane (79% purity, 811 mg, ca. 1.6 mmol, 2 eq.), tris(dibenzylideneacetone)dipalladium(0) (58.8 mg, ca. 4 mol %), tri(o-tolyl)phosphine (79.3 mg, ca. 16 mol%), and BrFBTT-Boc (382.3 mg, ca. 0.8 mmol) were mixed in a 50 mL flame-dried round bottom flask. After protection with argon, anhydrous toluene (16 mL, 0.1 M) was added via syringe. The mixture was refluxed overnight at 105 °C. Purification proceeded similarly to BrFBTT-Boc. The desired product was isolated with 100% yield. $^1$H NMR (400 MHz, CDCl$_3$) δ 8.16 (d, J = 3.1 Hz, 1H), 7.55 (d, J = 5.1 Hz, 1H), 7.15 (d, J = 5.3 Hz, 1H), 7.02 (d, J = 3.3 Hz, 1H), 4.74 (s, 1H), 3.53 – 3.46 (m, 2H), 3.13 (t, J = 6.7 Hz, 2H), 2.54 (q, J = 6.8 Hz, 2H), 1.46 (s, 9H), 1.20 (t, J = 7.7 Hz, 3H).

*TeFBTT, iodide salt, 2-(5-(7-(3-ethylthiophen-2-yl)-5,6-difluorobenzo[c][1,2,5]thiadiazol-4-yl)thiophen-2-yl)ethan-1-aminium iodide*: TeFBTT-Boc (391.9 mg, 0.7720 mmol, 1 equiv.) was dissolved in anhydrous ethanol (15.44 mL, 20 mL mmol$^{-1}$) (a few drops of dichloromethane were added to aid solubility), and 57 wt% aqueous HI solution (346.49 mg, 2.0 equiv.) was then added to cleave the boc protecting group and in situ from the ammonium iodides of the ligands. After protection with argon, the mixture was refluxed at 70 °C for 3 h and left stirring overnight. After the reaction, the solvent was mostly removed with rotary evaporation at 40 °C. Then the resulting crude product was reprecipitated at least thrice using ethanol and ether. The product was a bright yellow powder (80.9% yield). $^1$H NMR (500 MHz, DMSO-d$_6$) δ 8.15 (d, J = 3.8 Hz, 1H), 7.82 (d, J = 5.2 Hz, 1H), 7.79 (s, 3H), 7.23 (d, J = 5.2 Hz, 2H), 3.21 (q, J = 3.9 Hz, 4H), 2.48 (t, 2H), 1.13 (t, J = 7.6 Hz, 3H). $^{19}$F NMR (470 MHz, DMSO-d$_6$, uncalibrated) δ -128.41 (d, J = 19.9 Hz), -129.78 (d, J = 19.1 Hz). $^{13}$C NMR (126 MHz, DMSO-d$_6$) δ 150.47 (d, J = 7.7 Hz), 150.47 (dd, J = 209.7, 19.2 Hz), 148.44 (dd, $J_{CF}$ = 214.2, 19.2 Hz), 147.89 (d, $J_{CF}$ = 9.0 Hz), 145.00, 143.51 (d, $J_{CF}$ = 6.2 Hz), 131.59 (d, $J_{CF}$ = 7.9 Hz), 129.26, 128.47, 128.42, 126.86, 122.68, 112.50 (d, $J_{CF}$ = 12.4 Hz), 111.02 (d, $J_{CF}$ = 16.4 Hz), 27.43 (s, 2 C), 22.15, 14.59. See Supplementary Information for a detailed analysis of C−F coupling effects in $^{13}$C NMR spectra.

## Nuclear magnetic resonance

NMR spectra were acquired at room temperature with a Bruker Avance III HD 400 MHz spectrometer equipped with a 2-channel Nanobay console and a 5 mm BBFO Z-gradient SmartProbe, or Bruker NEO 500 MHz equipped with prodigy liquid-nitrogen cooled cryoprobe with BBFO configuration. NMR spectra were analyzed with Mestrenova. $^1$H peaks were calibrated with that of CHCl$_3$ (δ 7.26) in CDCl$_3$ or DMSO (δ 2.5) in DMSO-d$_6$. $^{13}$C peaks were calibrated with that of DMSO (δ 39.2) in DMSO-d$_6$. $^{19}$F peaks were uncalibrated. Processed NMR spectra are included in the Supplementary Information.

## Perovskite single-crystal growth

Bulk iodide perovskite single crystals were obtained through slow-cooling a solution composed of PbI$_2$ (10 mg) and the organic ammonium salt, TeFBTT (1 mg) in HI (200 μL), H$_3$PO$_2$ (100 μL), and ethanol (400 μL) as solvents. Ethanol increases the solubility of conjugated organic ammonium salts in aqueous condition reported in our previous study[16]. In detail, the precursors and solvents were mixed in a 4 mL scintillation vial. After being tightly capped, the sample vial was heated carefully in a boiling water bath until materials were completely dissolved, and the solution was clear. This process was carried out with great caution in a fume hood with sash closed. Sonication was occasionally used to assist the dissolution of materials, but magnetic stirring was avoided. Then, the vial was transferred to and sealed in a Dewar flask containing boiling water, which was then stored at ambient conditions to allow slow cooling for 72–96 h until the water bath reached room temperature. The crystal was stored in solution until being removed for structure determination and face indexing.

## Single crystal X-ray diffraction

Single crystals of the investigated compounds were coated with a trace of Fomblin oil and were transferred to the goniometer head of a Bruker Quest diffractometer. Data for (TeFBTT)$_2$PbI$_4$ were collected on an instrument with kappa geometry, a Cu Kα wavelength (λ = 1.54178 Å) I-μ-S microsource X-ray tube, a laterally graded multilayer (Goebel) mirror for monochromatization, and a Photon III C14 area detector. The instrument was equipped with an Oxford Cryosystems low-temperature device and examination and data collection were performed at 150 K.

Data were collected, reflections were indexed and processed, and the files scaled and corrected for absorption using APEX4[17] and SADABS or TWINABS[18]. The space groups were assigned using XPREP within the SHELXTL suite of programs[19,20] and solved by dual methods using ShelXT[21] and refined by full matrix least squares against F2 with all reflections using Shelxl2018 or Shelxl2019[22], and the graphical interface Shelxle[23]. Where not specified otherwise (Supplementary Table 2 and associated text), H atoms were handled as follows: H atoms attached to carbon and nitrogen atoms as well as hydroxyl hydrogens were positioned geometrically and constrained to ride on their parent atoms. C-H bond distances were constrained to 0.95 Å for aromatic and alkene C-H moieties, and to 0.99 and 0.98 Å for aliphatic CH$_2$ and CH$_3$ moieties, respectively. Methyl CH$_3$, ammonium NH$_3^+$ and hydroxyl H atoms were allowed to rotate but not to tip to best fit the experimental electron density. Water H atom positions were refined, and O-H distances were restrained to 0.84(2) Å. Where necessary, water H⋯H distances were restrained to 1.36(2) Å, and H atom positions were further restrained based on hydrogen bonding considerations. Uiso(H) values were set to a multiple of Ueq(C) with 1.5 for CH$_3$, NH$_3^+$ and OH, and 1.2 for C-H and CH$_2$ units, respectively.

Additional data collection and refinement details, including a description of the disorder (where present) can be found in Supplementary Table 2. Complete crystallographic data, in CIF format, have been deposited with the Cambridge Crystallographic Data Centre (CCDC), with the deposition number 2305444. CCDC contains the

supplementary crystallographic data for this paper. These data can be obtained free of charge from The Cambridge Crystallographic Data Centre via www.ccdc.cam.ac.uk/data_request/cif. The program Mercury was used for crystal structure illustration in this article[24].

### Device fabrication

The ITO-coated glass substrates were sequentially cleaned in detergent (Alconox), deionized water, acetone, and IPA using sonication for 15 min per step. After cleaning, the substrates underwent UV-ozone treatment for 20 min. The ZnO NCs were then applied via spin-coating at 2000 rpm for 30 s, followed by annealing at 150 °C for 10 min. A PEIE layer (0.04 wt% in IPA) was subsequently spin-coated at 5000 rpm and annealed at 100 °C for 10 min. The substrates were then moved into a nitrogen-filled glove box. Perovskite precursors were filtered using a 0.45 µm polytetrafluoroethylene (PTFE) syringe filter and spin-coated onto the ZnO/PEIE-coated substrates at 3000 rpm for 1 min, followed by annealing at 100 °C for 16 min. For samples treated with TeFBTT, a 50 µL solution of TeFBTT (dissolved in IPA and chlorobenzene in a 1:3 ratio) was spin-coated dynamically at 3000 rpm for 30 s. It is advised to prepare fresh TeFBTT solution before each use to prevent degradation in the solvent. Following this, a TFB layer (12 mg mL$^{-1}$ in toluene) was spin-coated at 4000 rpm. Finally, a 10 nm layer of MoO$_x$ and a 60 nm layer of Au were deposited sequentially using thermal evaporation under high vacuum (<1 × 10$^{-6}$ mbar). The device's active area was defined as 0.04 cm$^2$, determined by the overlap between the ITO and Au electrodes.

### Device characterizations

All LED devices were tested at room temperature inside a nitrogen-filled glove box without encapsulation. A Keithley 2450 source-measure unit was used to measure the current density versus voltage characteristics. Luminance properties, including radiance, EL, EQE, and operational stability, were evaluated using a 100 mm PTFE integrating sphere connected to a spectrometer (Enli Technology, LQ-100X). The system was calibrated using a NIST-traceable standard quartz tungsten halogen (QTH) lamp. Operational stability tests for the LEDs were conducted under constant current densities.

### Device cross-validation

LED devices were cross-validated at Princeton University in Prof. Barry Rand's group. Samples were initially sealed in a nitrogen-filled bag and then placed in an airtight shipping tube to minimize exposure to oxygen and moisture. Characterization was carried out at room temperature within a nitrogen-filled glove box without encapsulation. Angle-integrated emission measurements were performed using a custom-built goniometer setup equipped with a silicon photodiode calibrated with a known responsivity curve. $J$–$V$ measurements were taken using a Keithley 2400 SourceMeter, and emission properties were analyzed by collecting photocurrent in the forward-emitting direction at each voltage step with a Hewlett Packard 4140B pA Meter/DC Voltage Source. The emission spectrum was measured using a StellarNet EPP2000 spectrometer. Angular emission dependence was assessed by running two representative high-performing LEDs on each 3 × 3 cm$^2$ substrate at a constant current of 0.6 mA, with photocurrent collected at 10° intervals from 0° to 80° relative to the surface normal. The photodiode position was controlled by a ThorLabs TDC001 servo controller. The results from the two devices were compared with each other and against a Lambertian profile to ensure consistency and reliability. Based on these measurements, the forward emission photocurrent (at 0°) was used to estimate the photocurrent profile over the entire hemispherical emission range, assuming axial symmetry of the LED. This integration provided the total photon emission, with the emission spectrum determined to be angle-independent. From the collected data, EQE and radiance were calculated at each voltage step beyond turn-on.

### Morphological characterizations

The SEM images were acquired with a Hitachi S-4800 field emission SEM, while AFM measurements were performed using an Asylum Cypher ES AFM. Topographic AFM images were utilized to determine the average grain height ($H$) of the perovskites and the convex height ($h_s$) on the subsequently fabricated TFB layers. SEM topographic images were used to extract the average grain size and packing density ($\alpha$) of the grains. Further details can be found in Supplementary Note 2 (Supplementary Figs. 23–25). Cross-sectional specimens for TEM were prepared using a Thermo Scientific Helios G4 UX Dual Beam, and the TEM images along with EDS mapping were obtained with a Thermo Scientific Themis Z operating at an accelerating voltage of 80 kV.

### X-ray diffraction characterizations

XRD spectra were obtained by Rigaku Smart Lab using Cu K$\alpha$ source.

### Steady-state optical measurements

UV-Vis absorption spectra were measured using an Agilent Cary-5000 UV-Vis-NIR spectrometer in transmission mode. Steady-state PL spectra were recorded with an Olympus BX53 microscope system equipped with an X-CITE 120Q UV lamp. The microscope's filter cube included a 330–385 nm bandpass filter for excitation and a dichroic mirror with a 420 nm cutoff for light splitting. PL spectra were captured using a SpectraPro HRS-300 spectrometer. PLQY measurements were conducted in ambient air using a custom-designed system comprising a continuous wave OBIS 375 nm laser excitation source, an integrating sphere, an optical fiber, and the aforementioned spectrometer. The spectrometer was calibrated for intensity using a standard tungsten-halogen lamp (StellarNet SL1CAL) traceable to NIST standards. Perovskite films for PLQY analysis were fabricated on ITO/ZnO/PEIE substrates following the same method described for device fabrication. The laser power density details for PLQY measurements are presented in Supplementary Note 3 (Supplementary Fig. 26). To simulate the TFB layer and protect the films from oxygen and moisture, an optically inert isotactic poly(methyl methacrylate) (>80% isotactic, Sigma Aldrich) layer was spin-coated onto the perovskite (1 wt% solution in anhydrous toluene, spun at 4000 rpm for 30 s). This additional polymer layer was crucial for achieving significant PLQY. All PLQY measurements were performed immediately after film preparation.

### Nanoscale charge-carrier recombination rate mapping

Local transient photovoltage (TPV) decays in contact mode were measured using an Agilent 5500 conductive atomic force microscope. The conductive tip, a platinum/chromium-coated silicon model with a force constant of 0.20 N m$^{-1}$ and a resonance frequency of 14 kHz, was sourced from Budget Sensors. An Agilent MSOX4154A oscilloscope captured the transient TPV data, with an MGL-I-532 DPSS green laser (532 nm wavelength) serving as the light source to generate charge carriers. The oscilloscope received the transient signals from the AFM controller through an Agilent N9447A Breakout Box, and an input impedance of 1.0 MΩ was used to collect the transient decays. To determine the charge carrier recombination lifetime ($\tau_r$), a mono-exponential decay model was applied:

$$A(t) = A_0 e^{\frac{-t}{\tau}}$$

where $A_0$ is the steady-state photovoltage and $\tau$ is the time constant that represents charge carrier lifetime.

### Grazing incidence wide-angle X-ray scattering measurement

GIWAXS spectra were collected at beamline 7.3.3 of the Advanced Light Source at Lawrence Berkeley National Laboratory. Measurements were performed using an incident angle of 0.1° (or 0.1–0.4° for 2D/3D phase ratio analysis) and a wavelength of 1.24 Å (corresponding to 10 keV energy). The Pilatus 2 M detector (Dectris, Inc.) was used, and

data calibration was performed with silver behenate as a standard via the Igor Pro NIKA package. Perovskite films for GIWAXS analysis were prepared on Si/ZnO/PEIE substrates following the same method as described for device fabrication.

## Ultraviolet photoelectron spectroscopy

Ultraviolet photoelectron spectroscopy (UPS) was carried out using a PHI 5600 analysis system equipped with a hemispherical electron energy analyzer and a multichannel plate detector. The spectra were acquired using an H Lyman-α photon source (E-LUX 121) with a photon energy of 10.2 eV and a pass energy of 5.85 eV.

## Optical simulations

Please refer to Supplementary Note 4 (Supplementary Figs. 27–29 and Supplementary Tables 3 and 4) for detailed information on the simulation method.

A single cell (length: $P$) contains one perovskite grain simplified as a tetragonal block embedded in TFB with defined length ($l$) and height ($H$). These parameters were extracted from SEM and AFM as mentioned in Supplementary Note 2. The curvature atop TFB was simplified as a convex dome defined by the convex height ($h_s$). The thickness of TFB (50 nm) in the model refers to the distance from the top of perovskite grains to the top of the convex dome atop TFB. Gap between the perovskite grains was assumed to be filled with TFB. Refractive indexes were extracted from the ellipsometer measurement (Supplementary Fig. 27). We chose 800 nm as the wavelength of interest. Note the imaginary part of the refractive index of perovskite is ignored in the simulation.

It is important to clarify that the perovskite block in the cell is the only dipole source (i.e., the emitter). Also, no periodic boundary was used in our simulation. Hence, if the simulation scale extends beyond a single cell, blocks nearby are considered passive structures, while the dipole source is placed as close as possible to the center. Multiple simulations with different polarizations were carried out to simulate the incoherent isotropic light source in a 4×4 uniformly distributed dipole source in one perovskite region.

Localized refined meshes ($\Delta x$, $\Delta y$, and $\Delta z$) were applied. A mesh size of 2 nm is applied in all three directions in the dipole source region and $\Delta z = 1$ nm in the convex structures. A perfect match layer boundary condition is used at the glass substrate and all $x$, $y$ directions. A metal boundary condition is applied at the gold layer. Lumerical's far-field analysis group is implemented to calculate the far-field out-coupling power using the near-field power at the glass-ITO interface. The theoretical OCE is then calculated with the division of outcoupling power by the source power obtained inside the dipole power box.

## Molecular simulations

Geometry optimizations and excited state calculations for TeFBTT were carried out by means of Density Function Theory (DFT) and Time-Dependent DFT (TDDFT) as implemented in the Gaussian 16 package[25] using the B3LYP[26,27] functional and def2-TZVP basis set in vacuum. An alkylamine (·CH₂-CH₂-NH₂) was used instead of the alkylammonium tail. Based on our experience, expensive basis set like def2-TZVP is necessary to achieve adequate accuracy in DFT and TDDFT calculations when considering the band alignment within perovskites.

## Data availability

Crystallographic data for (TeFBTT)₂PbI₄ have been deposited in the Cambridge Crystallographic Data Centre (CCDC) database, with reference number 2305444. Other data can be found in the main text or the Supplementary Information. Source data are provided with this paper.

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

## Acknowledgements

This work is primarily supported by the U.S. National Science Foundation (Grants No. 2131608-ECCS and 2143568-DMR), U.S. Department of Energy, Office of Science, Office of Basic Energy Sciences (Grant No. DE-SC0022082), U.S. Department of Energy's Office of Energy Efficiency and Renewable Energy (EERE) under the Solar Energy Technologies Office (Grant No. DE-EE0009519). S.-D.B. acknowledges support from the Basic Science Research Program through the National Research Foundation of Korea (NRF) funded by the Ministry of Education (Grants No. 2021R1I1A1A01048035). K.R.G. acknowledges support from the U.S. National Science Foundation (Grant No. 2102257-DMR). The authors acknowledge Sarah N. Chowdhury and Alexandra Boltasseva (Purdue University) for refractive index data of each layer in the device by ellipsometry. The authors acknowledge Wenjing Li for the automated analysis of PLQY data. The authors acknowledge Qixuan Hu for GC-MS data. The views expressed herein do not necessarily represent the views of the U.S. Department of Energy or the United States Government.

## Author contributions

S.-D.B., W.S., and L.D. conceived the work. S.-D.B. developed high-performance PeLEDs. W.S. designed and synthesized the TeFBTT organic cation. W.F. and L.J.G. performed outcoupling simulations. Y.T. and Y.H.L. inspired the solvent-engineering approach. J.L., W.B.G., and B.P.R. performed the device cross-validation. Y.Z., M.B.F.S., P.I.K., and Q.Q. performed nanoscale charge-carrier recombination mapping. A.H.C. and C.Z. performed GIWAXS measurements. J.Y.P. synthesized the (TeFBTT)$_2$PbI$_4$ single crystal. H.R.A. and K.R.G. performed UPS. H.Y., K.W., S.J.Y., Y.-T.Y., J.Q., and F.G. assisted in interpreting the results. L.D. supervised the work. S.-D.B. drafted the manuscript, with assistance from W.S. S.-D.B., W.S., and L.D. revised the manuscript. All authors discussed the results and commented on the manuscript.

## Competing interests

The authors declare no competing interests.
