## [Transparent Peer Review file · Nature Communications]

Grain engineering for efficient near-infrared perovskite light-emitting diodes

Corresponding Author: Professor Letian Dou

Version 0:

Reviewer comments:

Reviewer #1

(Remarks to the Author)

The authors have made considerable efforts to address all major concerns raised in the prior round of review. The revision has significantly improved the manuscript, particularly in terms of novelty statement, clarity in discussion, and methodological detail. Overall, the authors have demonstrated a strong commitment to addressing the reviewers' concerns, and their revisions have significantly enhanced the manuscript's quality. I have no further major concerns and believe the manuscript is now suitable for publication in its current form.

Reviewer #2

(Remarks to the Author)

The authors have made necessary revisions following the comments, and I would like to recommend acceptance of the revised paper for publication in Nature Communications.

Reviewer #3

(Remarks to the Author)

The paper has significantly improved, and the authors have addressed most of the comments appropriately. Just a minor point: the authors should provide a more extensive explanation of why the perovskite thin films in Figure 1 (or Figure R1) do not form a conformal layer, which is essential for LED devices. A discussion on this aspect would further enhance the paper.

Response to Reviewer #1

The authors have made considerable efforts to address all major concerns raised in the prior round of review. The revision has significantly improved the manuscript, particularly in terms of novelty statement, clarity in discussion, and methodological detail. Overall, the authors have demonstrated a strong commitment to addressing the reviewers' concerns, and their revisions have significantly enhanced the manuscript's quality. I have no further major concerns and believe the manuscript is now suitable for publication in its current form.

Response: Thank you for your positive feedback and for recognizing our efforts to improve the manuscript. We are grateful for the time and consideration you have dedicated to reviewing our submission. Thank you once again for your support.

Response to Reviewer #2

The authors have made necessary revisions following the comments, and I would like to recommend acceptance of the revised paper for publication in Nature Communications.

Response: Thank you for your positive feedback and for recommending the acceptance of our revised manuscript for publication. We are grateful for the time and consideration you have dedicated to reviewing our submission. Thank you once again for your support.

Response to Reviewer #3

The paper has significantly improved, and the authors have addressed most of the comments appropriately. Just a minor point: the authors should provide a more extensive explanation of why the perovskite thin films in Figure 1 (or Figure R1) do not form a conformal layer, which is essential for LED devices. A discussion on this aspect would further enhance the paper

Response: Thank you once again for your valuable feedback and for recommending the acceptance of our revised manuscript for publication. We greatly appreciate your support and the time you have dedicated to reviewing our work.

In response to your latest comment regarding the formation of perovskite thin films in Figure 1, we have provided additional explanations in the revised manuscript.

“It is generally observed that amino acid additives – 5-aminovaleric acid (5AVA) in this case – can easily lead to the formation of discontinuous FAPbI₃ islands on a zinc oxide (ZnO)/polyethylenimine ethoxylated (PEIE) layer, primarily due to a dehydration reaction of 5AVA with ZnO-PEIE surface during annealing⁴. This reaction facilitates the formation of a FAPbI₃ island structure while concurrently generating thin organic layers that bridge voids between the discontinuous grains, thereby mitigating shunt current pathways. This phenomenon becomes more pronounced with the addition of NMP, primarily attributed to its discernibly higher boiling point compared with DMF, resulting in larger crystal grains.”